

# A bioinformatics approach to identifying *Wolbachia* infections in arthropods

Jane Pascar[1,2] and Christopher H. Chandler[1]

[1] Department of Biological Sciences, State University of New York at Oswego, Oswego, NY, United States of America
[2] Department of Biology, Syracuse University, Syracuse, NY, United States of America

## ABSTRACT

*Wolbachia* is the most widespread endosymbiont, infecting >20% of arthropod species, and capable of drastically manipulating the host's reproductive mechanisms. Conventionally, diagnosis has relied on PCR amplification; however, PCR is not always a reliable diagnostic technique due to primer specificity, strain diversity, degree of infection and/or tissue sampled. Here, we look for evidence of *Wolbachia* infection across a wide array of arthropod species using a bioinformatic approach to detect the *Wolbachia* genes *ftsZ, wsp,* and the *groE* operon in next-generation sequencing samples available through the NCBI Sequence Read Archive. For samples showing signs of infection, we attempted to assemble entire *Wolbachia* genomes, and in order to better understand the relationships between hosts and symbionts, phylogenies were constructed using the assembled gene sequences. Out of the 34 species with positively identified infections, eight species of arthropod had not previously been recorded to harbor *Wolbachia* infection. All putative infections cluster with known representative strains belonging to supergroup A or B, which are known to only infect arthropods. This study presents an efficient bioinformatic approach for post-sequencing diagnosis and analysis of *Wolbachia* infection in arthropods.

## INTRODUCTION

Symbiotic relationships are ubiquitous in nature and can vary between parasitic, commensal, and mutalistic. *Wolbachia* is a diverse and widespread α-proteobacterium and obligatory endosymbiont (*Werren, 1997a*; *Werren, 1997b*; *Saridaki & Bourtzis, 2010*). *Wolbachia* was first described in *Culex pipiens* (*Hertig, 1936*) and has since been identified in various clades of arthropods including Chelicerata (*Werren & Windsor, 2000*) (*Mock et al., 2016*), Crustacea (*Bouchon, Rigaud & Juchault, 1998*; *Cordaux, Michel-Salzat & Bouchon, 2001*; *Cordaux et al., 2012*), and Hexapoda (*Werren & Windsor, 2000*; *Clark et al., 2001*; *Augustinos et al., 2011*; *Bing et al., 2014*). Conservative estimates suggest that the frequency of *Wolbachia* infection in arthropods is at least 20% (*Werren, 1997b*; *Werren & Windsor, 2000*), while one study suggests a prevalence as high as 76% of arthropod species (*Jeyaprakash & Hoy, 2000*). Meta-analysis indicates that the infection distribution in the total number of species may be closer to 66% (*Hilgenboecker et al., 2008*).

Corresponding author
Jane Pascar, jpascar@oswego.edu

*Wolbachia* is normally transmitted vertically, from mother to offspring, and can manipulate the host's reproduction through five mechanisms: cytoplasmic incompatibility, parthenogenesis, male killing, feminization (*Cordaux, Bouchon & Grève, 2011*) and meiotic drive (*Kageyama et al., 2017*). However, there is evidence that *Wolbachia* can be horizontally transmitted (*Vavre et al., 1999*; *Cordaux, Michel-Salzat & Bouchon, 2001*; *Raychoudhury et al., 2009*; *Kraaijeveld et al., 2011*). Recently, discrete reciprocal benefits provided by infection have been observed including a positive impact on host immunity (*Teixeira, Ferreira & Ashburner, 2008*; *Osborne et al., 2009*), immunocompetence (*Braquart-Varnier et al., 2008*), fecundity (*Weeks et al., 2007*), and metabolic activity (*Darby et al., 2012*).

Currently, all *Wolbachia* strains are classified as a single species, with further classification into at least sixteen supergroups, A–Q (*Lo et al., 2007*; *Lindsey et al., 2016a*). The four most well studied clades are supergroups A–D. Supergroups A and B are monophyletic and are the most common supergroups known to infect arthropods, while supergroups C and D infect filarial nematodes (*Gerth et al., 2014*). Supergroup G was discovered to be a recombinant between supergroups A and B; thus it is no longer considered a distinct lineage (*Baldo & Werren, 2007*). Supergroups E–Q infect a variety of hosts including nematodes, springtails, termites, fleas, aphids, and mites (*Lo et al., 2002*; *Casiraghi et al., 2005*; *Ros et al., 2009*; *Haegeman et al., 2009*; *Augustinos et al., 2011*; *Bing et al., 2014*; *Glowska et al., 2015*).

*Wolbachia* has a relatively small genome at about 0.9–1.5 Mbp. Historically, *Wolbachia* infection was diagnosed using 16S rRNA sequences; however, strains range in divergence from 0.2% to 2.6%, and when used independently, 16S provides limited information for inferring phylogenetic relationships (*O'Neill et al., 1992*). *wsp*, *ftsZ* and the *groE* operon are all protein-encoding genes used for the detection and phylogenetic analysis of *Wolbachia* (*Van Borm et al., 2003*). The *ftsZ* gene is involved in cell division and is highly conserved in unculturable bacteria species (*Holden, Brookfield & Jones, 1993*), but regions that are relatively higher in divergence make it a candidate for better phylogenetic resolution allowing the distinction between supergroups A and B to become apparent (*Werren, 1997a*). The *wsp* gene, which codes for the surface protein WSP in *Wolbachia*, shows an even higher variability and faster evolutionary rate than 16S or *ftsZ* and can be used in identifying groups and strains of *Wolbachia* (*Zhou, Rousset & O'Neil, 1998*; *Braig et al., 1998*), but also displays recombination, which can be misleading when used in phylogenetic analyses (*Baldo & Werren, 2007*). The *groE-* homologous operon has been noted as another candidate for resolving strain taxonomy (*Masui, Sasaki & Ishikawa, 1997*). Only a single copy of the operon exists in the genome and it includes the genes that encode the heat shock proteins GroES and GroEL, which are separated by a non-coding intergenic region that is thought to be faster evolving than either of the coding regions (*Masui, Sasaki & Ishikawa, 1997*).

With the use of antibiotics, *Wolbachia* infections in some species have been cured and the phenotypic changes that are induced by infection are consequently reversed (*Stouthamer, Luck & Hamilton, 1990*; *Bourtzis et al., 1994*; *Giordano, Jackson & Robertson, 1997*). More recently, *Wolbachia* has been proposed as a natural solution to controlling the spread of vector-borne diseases like malaria, yellow fever, and dengue (*Hoffmann et al., 2011*; *Walker & Moreira, 2011*; *Baldini et al., 2014*). Arthropods are present in nearly every habitat on

Earth and they play important ecological roles in a variety of niches. With an estimated 2.4–10.2 million species of arthropods (*Ødegaard, 2000*) it is important to quantify the prevalence and distribution of *Wolbachia* infection.

*Wolbachia* infections are typically diagnosed via polymerase chain reaction (PCR), using *Wolbachia*- specific primers. However, PCR-based tests may produce false positives or false negatives, depending on the strain of *Wolbachia* and the presence of other related bacterial symbionts (*Simões et al., 2011*). A metagenomics-based approach can also be useful for characterizing microbiomes, including looking for *Wolbachia* and other symbionts (e.g. *Dittmer & Bouchon, 2018*), and can even provide whole-genome sequence information for the symbiont (e.g., *Salzberg et al., 2005*; *Richardson et al., 2012*; *Saha et al., 2012*; *Campana, Robles García & Tuross, 2015*; *Derks et al., 2015*; *Wang & Chandler, 2016*; *Lindsey et al., 2016b*; *Gerth & Hurst, 2017*). While performing a high-throughput sequencing-based screen for *Wolbachia* involving hundreds of different species would require a huge sampling effort and could be cost-prohibitive, screening existing sequence datasets generated for other projects offers a powerful opportunity to diagnose novel infections and better characterize variation in symbionts.

Here, using publicly accessible next-generation sequencing data available in the NCBI Sequence Read Archive (SRA), we looked for evidence of *Wolbachia* infection in a diverse assemblage of arthropod species. We present methods for bioinformatically identifying *Wolbachia* infections in genomic samples. We then used these sequence data to assemble a draft genome sequence for each *Wolbachia* isolate and reconstruct the phylogenetic relationships among the identified *Wolbachia* strains. Using this approach, we uncover novel *Wolbachia* infections, as well as find possible evidence for horizontal transfer of *Wolbachia* between hosts and parasites. These results illustrate how existing genetic databases can provide a wealth of information on symbiotic microbes as a byproduct of host sequencing.

## MATERIALS & METHODS

### Retrieving data

All samples tested are available through the NCBI Sequence Read Archive (SRA) (Table S1). To identify samples for testing, all accession numbers that matched the criteria of Arthropoda genomic DNA were sent to the NCBI Run Selector (as of January 2017). In the Run Selector samples were selected based on the criteria that they were run on an Illumina platform, have a genomic library source, a random library selection and the library layout is paired. Transcriptome samples were excluded because of the possibility that some RNA preparation methods may select against bacterial RNA (e.g., poly-A enrichment *Westermann, Gorski & Vogel, 2012*) thus increasing the likelihood of false negatives and because assembling *Wolbachia* genomes would be impossible with these data. Similarly, targeted sequencing (e.g., RAD-seq) samples were excluded due to the possibility that the sequences used for detecting *Wolbachia* infections might be excluded during the library preparation process. Only paired-end datasets were considered in order to facilitate whole-genome assembly for positive samples, but there were relatively few species (only

22) in the database with single-end datasets that otherwise met our criteria. Every species that had a sample that met our criteria was chosen for sampling. Some species were over-represented in the number of runs that are available in the SRA; depending on the number of samples available in the SRA, an appropriate amount to include in our dataset was determined on a case-by-case basis (Table S2). Fastq-dump v. 2.8.0 from the SRA Toolkit (NCBI SRA) was used to download, at most, $5 \times 10^7$ reads from each accession.

## Diagnosing *Wolbachia* Infection

Magic-Blast v1.1.0 (NCBI) was used to compare the SRA reads to selected reference *wsp*, *ftsZ*, and *groE* operon sequences isolated from *Wolbachia* samples that are representative of supergroups A–D (Table 1). A custom R script identified SRA samples where there were matches at least 98 bp in length, ≥95% identity to one or more of the reference genes, and with three or more matching sequence reads. All samples that met these criteria were called *Wolbachia* positive samples.

To look for previous reports of *Wolbachia* infection in the species that tested positive, first Google Scholar was used. [species] + *Wolbachia* was used for the search terms. If no published results were found, next we used NCBI GenBank with the same search parameters to look for deposited sequences that may be unpublished that would indicate that *Wolbachia* had been found in the host species previously.

## Assembling the *Wolbachia* gene and genome sequences

From all the samples that tested positive (Table S3) if there were more than 3 samples from one species a maximum of 3 samples were chosen for downstream analysis (Table S4). Velvet v1.2.10 (*Zerbino & Birney, 2008*) was used to separately assemble the *wsp*, *ftsZ*, and *groE* sequences for each biological sample using the sequence reads that aligned to each gene in the previous step. It was run for kmer values of 21, 31, 41, and 51, using the automatic coverage cutoff flag. To select the optimal assembly of each gene for each sample, we performed BLASTn v2.28 (*Altschul et al., 1990*), which searched against a database made of each respective reference gene (Table 1). BEDTools v2.25.0 (*Quinlan & Hall, 2010*) and a custom script was used to parse the BLASTn results for the single longest contig matching each gene from each sample.

To assemble draft genomes for each *Wolbachia* isolate we identified, an iterative bait-and-assemble approach was used. Independent SRA experiments or runs from the same BioSample were first combined into a single dataset. For each sample, the mirabait tool from MIRA v4.0.2 (*Chevreux, Wetter & Suhai, 1999*) was then used to extract all reads from the full dataset that shared at least one kmer with at least one of seven reference *Wolbachia* genomes representing *Wolbachia* isolates from insects and nematodes (wPip, GCF_000073005.1; wMel, GCF_000008025.1; wNo, GCF_000376585.1; wRi, GCA_000022285.1; wVol, GCF_000530755.1; wCle, GCF_000829315.1; wTpre, GCF_001439985.1), using $k = 31$. These reads, and their corresponding paired-end partners, were assembled using SPAdes 3.11.1 (*Bankevich et al., 2012*). All resulting contigs were then aligned to the reference *Wolbachia* genomes using dc-megablast 2.7.0+ (*Camacho et al., 2009*), and any contig that matched any of the reference genomes with an e-value of

Pascar and Chandler (2018), *PeerJ*, DOI 10.7717/peerj.5486

**Table 1** **Reference *Wolbachia* genes.** Gene sequences from *Wolbachia*-infected hosts used to create the reference database for Magic-BLAST searches of SRA accessions to diagnose novel *Wolbachia* infections.

| Phylum | Class | Order | Species | Supergroup | Strain | Accession number | Gene | Citation |
|---|---|---|---|---|---|---|---|---|
| | Arachnida | Trombidiformes | *Bryobia praetiosa* | B | – | JN572870.1 | *wsp* | *Ros et al. (2012)* |
| | | | *Diaea circumlita* | A | wDiacir3 | AY486091.1 | *wsp* | *Rowley, Raven & McGraw (2004)* |
| | | Coleoptera | *Tribolium confusum* | – | NFR114 | AB469356.1 | *wsp* | D Kageyama, S Narita, T Imamura and A Miyanoshita (2008, unpublished data) |
| | | | *Tribolium confusum* | – | – | DQ842337.1 | *ftsZ* | *Baldo et al. (2006)* |
| | | | *Dicladispa armigera* | A | wDic | DQ243935.1 | *groE* | *Wiwatanaratanabutr et al. (2009)* |
| | | Diptera | *Culex pipiens* | B | – | DQ900650.1 | *wsp* | ND Djadid, N Daneshinia, S Gholizadeh and S Zakeri (2006, unpublished data) |
| | | | *Culex quinquefasciatus* | B | – | AY462861.1 | *wsp* | *Tsai et al. (2004)* |
| | | | *Drosophila melanogaster* | A | wMel | FJ403332.1 | *wsp* | YF Wang and Y Zheng (2008, unpublished data) |
| | | | *Drosophila simulans* | A | wMel | DQ412101.1 | *wsp* | *Mateos et al. (2006)* |
| | | | *Protocalliphora sialia* | B | wProtPA | AF448376.1 | *wsp* | *Werren & Bartos (2001)* |
| | | | *Culex quinquefasciatus* | B | 22 | GU901159.1 | *ftsZ* | JC Rondan-Duenas, A Blanco, and CN Gardenal (2010, unpublished data) |
| | | | *Drosophila melanogaster* | A | Canton-S | X71906.1 | *ftsZ* | *Holden, Brookfield & Jones (1993)* |
| | | | *Drosophila recens* | A | – | U28174.1 | *ftsZ* | *Werren, Zhang & Guo (1995)* |
| | | | *Drosophila simulans* | A | wHa | AY508998.1 | *ftsZ* | JWO Ballard (2003, unpublished data) |
| | | | *Drosophila simulans* | A | wMa(Ma) | AY508999.1 | *ftsZ* | JWO Ballard (2003, unpublished data) |
| | | | *Aedes albopictus* | A | wAlbA | DQ243927.1 | *groE* | *Wiwatanaratanabutr et al. (2009)* |
| | | | *Culex fuscocephala* | B | wFusc | AJ511284.1 | *groE* | S Wiwatanaratanabutr and P Kittayapong (2002, unpublished data) |
| | | | *Drosophila simulans* | A | – | AB002287.1 | *groE* | *Masui, Sasaki & Ishikawa (1997)* |
| | | | *Drosophila simulans* | A | – | AB002288.1 | *groE* | *Masui, Sasaki & Ishikawa (1997)* |
| | | | *Drosophila tristis* | A | – | AY563553.1 | *groE* | *Haine, Pickup & Cook (2005)* |

**Host Classification**

(*continued on next page*)

**Table 1** (*continued*)

| Phylum | Class | Order | Species | Supergroup | Strain | Accession number | Gene | Citation |
|--------|-------|-------|---------|------------|--------|------------------|------|----------|
| Arthropoda | | | *Encarsia formosa* | B | – | KC161951.1 | *wsp* | F Lu and MX Jiang (2012, unpublished data) |
| | Insecta | | *Muscidifurax uniraptor* | A | – | DQ380857.1 | *wsp* | *Kyei-Poku et al. (2006)* |
| | | | *Nasonia giraulti* | – | wNGirVA | AF448381.1 | *wsp* | *Werren & Bartos (2001)* |
| | | | *Nasonia vitripennis* | – | wNvi-2 | KC161919.1 | *wsp* | F Lu and MX Jiang (2012, unpublished data) |
| | | Hymenoptera | *Trichogramma cordubensis* | B | Sib | AF245164.1 | *wsp* | *Pintureau et al. (2000)* |
| | | | *Diplolepis rosae* | B | Type I | U83887.1 | *ftsZ* | *Schilthuizen & Stouthamer, 1998* |
| | | | *Habrocytus bedeguaris* | B | Type I | U83886.1 | *ftsZ* | *Schilthuizen & Stouthamer, 1998* |
| | | | *Trichogramma* n. spec. (nr. deion) | B | – | U59696.1 | *ftsZ* | *Schilthuizen & Stouthamer, 1998* |
| | | | *Asobara tabida* | – | – | AJ634749.1 | *groE* | *Haine, Pickup & Cook (2005)* |
| | | | *Bombyx mandarina* | – | – | KJ659909.1 | *wsp* | *Zha et al. (2014)* |
| | | | *Acraea encedon* | B | – | AJ130892.1 | *ftsZ* | *Hurst & Jiggins (1999)* |
| | | | *Ephestia kuehniella* | A | Type II | U62126.1 | *ftsZ* | *Schilthuizen, Honda & Stouthamer (1998)* |
| | | Lepidoptera | *Acraea pharsalus* | B | – | AJ318481.1 | *groE* | *Jiggins et al. (2002)* |
| | | | *Ephestia cautella* | A | – | AB002289.1 | *groE* | *Masui, Sasaki & Ishikawa (1997)* |
| | | | *Ephestia cautella* | B | – | AB002290.1 | *groE* | *Masui, Sasaki & Ishikawa (1997)* |
| | | Orthoptera | *Gryllus pennsylvanicus* | B | – | U28195.1 | *ftsZ* | *Werren, Windsor & Guo (1995)* |
| | Malacostraca | Isopoda | *Chaetophiloscia elongata* | – | – | AM087239.1 | *groE* | *Wiwatanaratanabutr et al. (2009)* |

Pascar and Chandler (2018), *PeerJ*, DOI 10.7717/peerj.5486

**Table 1** (*continued*)

| | Host Classification | | | | | | | |
|---|---|---|---|---|---|---|---|---|
| **Phylum** | **Class** | **Order** | **Species** | **Supergroup** | **Strain** | **Accession number** | **Gene** | **Citation** |
| | | | *Dirofilaria repens* | C | – | AJ252176.1 | *wsp* | *Bazzocchi et al. (2000)* |
| | | | *Litomosoides sigmodontis* | D | – | AJ252177.1 | *wsp* | *Bazzocchi et al. (2000)* |
| | | | *Onchocerca gibsoni* | C | – | AJ252178.1 | *wsp* | *Bazzocchi et al. (2000)* |
| Nematoda | Secementea | Spirurida | *Brugia malayi* | D | – | AJ252061.1 | *wsp* | *Bazzocchi et al. (2000)* |
| | | | *Brugia pahangi* | D | – | AJ252175.1 | *wsp* | *Bazzocchi et al. (2000)* |
| | | | *Dirofilaria immitis* | C | – | AJ252062.1 | *wsp* | *Bazzocchi et al. (2000)* |
| | | | *Wuchereria bancrofti* | D | – | AF285273.1 | *groE* | S Salahuddeen and TB Nutman (2000, unpublished data) |

**Table 2 *Wolbachia* sequences of known origin for phylogenetic analysis.** *Wolbachia* genes used as controls and the species name from which they were isolated. The supergroup of the *Wolbachia* strain is listed and these genes served as a control during the creation of the phylogeny.

| Host Classification | | | Gene & Accession Number | | | |
|---|---|---|---|---|---|---|
| **Phylum** | **Order** | **Species** | *ftsZ* | *groEL/groES* | **Supergroup** | **Citation** |
| Arthropoda | Isopoda | *Armidillidium vulgare* | DQ778101 | DQ778104 | B | *Verne et al. (2007)* |
| | Hemiptera | *Bemisia afer* | KF452573 | KF452533 | B | *Bing et al. (2014)* |
| | Hemiptera | *Bemisia tabaci* | KF452577 | KF452536 | B | *Bing et al. (2014)* |
| | Diptera | *Drosophila ambigua* | AY563550 | AY563552 | A | *Haine, Pickup & Cook (2005)* |
| | Diptera | *Drosophila melanogaster* | DQ235339 | DQ235379 | A | *Paraskevopoulos et al. (2006)* |
| | Diptera | *Drosophila tristis* | AY563551 | AY563553 | A | *Haine, Pickup & Cook (2005)* |

$10^{-10}$ or better, alignment length of at least 100 bp, and percent identity of at least 70%, was retained. This process was then repeated for a total of five iterations, in each cycle using mirabait to identify reads sharing one or more kmers with the last set of assembled contigs, re-assembling these putatively *Wolbachia*-derived reads, and retaining any of the newly assembled contigs that show similarity to a *Wolbachia* reference genome in BLAST searches. The quality of each final assembly was evaluated using QUAST v4.4 (*Gurevich et al., 2013*) and BUSCO v3.0.2b (*Simão et al., 2015*) with the Bacteria *odb9* reference gene set. Finally, we mapped all sequencing reads from each associated BioSample (not just those used for the assembly process) to the corresponding assembly using bwa mem v.0.7.17 (*Li, 2013*), and then used the sambamba depth command (*Tarasov et al., 2015*) to extract coverage information for each assembled contig over 400 bp in length, excluding 150 bp from the ends of the contigs (where coverage tends to drop off because reads extending beyond the contig may fail to map successfully).

## Phylogenetic analysis

We first constructed phylogenies using the assembled *ftsZ* and *groE* sequences, as well as from a concatenated dataset of both genes; *wsp* was excluded from phylogenetic analyses because of its high frequency of recombination (*Baldo & Werren, 2007*). *Wolbachia* gene sequences representing *ftsZ* and the *groE* operon from other studies where the supergroup classification was determined were used as control samples; in this analysis, we included only reference sequences where both genes had been sequenced from the same biological sample (Table 2). MAFFT v7.310 (*Katoh & Standley, 2013*) was used to align the sequences for each respective gene. Samples that lacked sufficient length of matching base pairs (at least 800 bp in total across both genes) were discarded from downstream analysis. GBLOCKS v0.91b (*Castresana, 2000*) removed the poorly aligned portions of the sequences from each gene alignment using the default parameters. MEGA v7.0 (*Kumar, Stecher & Tamura, 2016*) was used to construct phylogenies using maximum likelihood. The model for which the phylogenies were constructed was chosen according to MEGA's suggestion for best fit based on the lowest Bayesian information criterion (BIC) (Table S5). Node support was assessed by bootstrapping with 1,000 replicates.

We also constructed phylogenies based on whole-genome data from a subset of the assemblies which appeared the most complete based on the BUSCO assessment. For

these phylogenies, we used REALPHY (*Bertels et al., 2014*), to align genome sequences and identify loci for inclusion in the phylogenetic analysis, using as a reference the seven *Wolbachia* genomes used in the assembly process and merging the reference alignments with the default parameters. We then performed phylogenetic analysis by maximum likelihood in RAxML v8.2.11 (*Stamatakis, 2014*) using the TVM+I+G model as selected by ModelTest-NG v0.1.2 (the successor to jModelTest); (*Darriba et al., 2012*) using AIC. The RAxML analysis included 100 independent replicate searches for the best-scoring tree and 200 bootstrap replicates to assess node support.

## RESULTS

### Diagnosing *Wolbachia* infections from publicly available sequence data

A total of 2,545 individual 'runs' from the SRA, representing 288 species and subspecies were tested for *Wolbachia* (Table S1). Of those, 173 runs from 34 unique species tested positive for the selected reference *Wolbachia* genes (Table S3). That is, 11.8% of species tested positive for *Wolbachia* in at least one sample and only 6.8% of all SRA runs tested positive. All samples that tested positive were from samples that are in the class Insecta and representative of five orders: Coleoptera, Diptera, Hymenoptera, Hemiptera, and Lepidoptera. According to our literature search eight of these species have not previously been confirmed to have *Wolbachia* infections—*Bembidion lapponicum*, *Ceratina calcarata*, *Delias oraia*, *Diachasma alloeum*, *Diploeciton nevermanni*, *Ecitophyla simulans*, *Gerris buenoi* and *Isocolus centaureae* (Table 3).

### Assembling *Wolbachia* genomes

In total, we assembled draft genomes for 51 *Wolbachia* isolates (Table 4), including at least one for each of the 34 unique host species. There were only two cases in which the assembly was substantially smaller than the expected genome size. In one of those (*Biorhiza pallida* 3), infection was confirmed in independent biological samples, and in the other (*Mycopsylla proxima*) the small assembly probably resulted from the small size of the input dataset. The rest of these assemblies appeared nearly complete, with total assembly sizes of at least 1 Mb and high numbers of BUSCO reference genes represented by a single gene in the assembly. All assemblies were missing at least 13 of the BUSCO reference genes.

We also sought to determine whether each sample that tested positive was likely to represent an actual *Wolbachia* infection, or the result of *Wolbachia* sequences horizontally transferred into the host genome. If the sequencing depth of the *Wolbachia*-like contigs in the assembly differs substantially from the sequencing depth of the host genome, then horizontal transfer can be ruled out. However, performing whole-genome assembly with every sequence dataset to estimate the sequencing depth of the host genome was computationally time-consuming, and our attempts to estimate sequencing depth more rapidly by counting $k$-mers in the raw data were unsuccessful in most cases because of low sequencing depth. Therefore, we obtained estimates of the genome size of host species from other sources, such as draft assemblies available at NCBI (Table 4), when available; although draft assemblies can differ substantially in size from actual genome sizes, for

Pascar and Chandler (2018), *PeerJ*, DOI 10.7717/peerj.5486

**Table 3  Species showing evidence of *Wolbachia* infection.**  List of unique species (class; order) that tested positive for the presence of *Wolbachia* genes.

| Phylum | Class | Order | Species | Supergroup (this study) | Supergroup (previous study) | Citation | GenBank Accession numbers |
|---|---|---|---|---|---|---|---|
| | | | *Bembidion lapponicum | B | – | – | – |
| | | | Callosobruchus chinensis | A (but possible double infection) | A/B | Kondo, Shimada & Fukatsu (1999), Kondo et al. (2002), D Kageyama, S Narita, T Ima­mura and A Miyanoshita (2008, unpublished data) | AB025965; AB080664, AB080665, and AB081842; AB469358 |
| | | | Diabrotica virgifera virgifera | A | A | Giordano, Jackson & Robertson (1997); R Giordano, L Clark, R Alvarez-Zagoya and JF Perez-Dominguez (2005, unpub­lished data) | U83098, AF011270–AF011271; DQ091306, DQ091307, DQ091308, DQ091309 |
| | | Coleoptera | *Diploeciton nevermanni | B | – | – | – |
| | | | *Ecitophya simulans | B | – | – | – |
| | | | Anopheles gambiae | B | A | Baldini et al. (2014) | KJ728739–KJ728755 |
| | | | Drosophila melanogaster | A | A | Bourtzis et al. (1994), Woolfit et al. (2013) | Z28981 Z28982 Z28983; KI440871–KI440895 |
| | | | Drosophila simulans | A/B | A/B | Riegler et al. (2004); Ellegaard et al. (2013) | AY227739, AY227742; CP003883, CP003884 |
| | | | Drosophila triauraria | A | A | Cordaux et al. (2008) | EU714523 |
| | | Diptera | Drosophila yakuba | A | A | Charlat, Ballard & Mercot (2004), Zabalou et al. (2004), Ioannidis et al. (2007), Cordaux et al. (2008) | AY291346, AY291348; AJ620679; DQ498875; EU714519 |
| | | | Rhagoletis pomonella | A | A | Schuler et al. (2011) | HQ333145, HQ333146, HQ333147, HQ333148, HQ333149, HQ333150, HQ333151, HQ333152, HQ333153, HQ333154, HQ333155, HQ333156, HQ333157, HQ333158, HQ333159 |
| | | | Rhagoletis zephyria | A | A | Schuler et al. (2011) | – |

**Table 3** (*continued*)

| Phylum | Class | Order | Species | Supergroup (this study) | Supergroup (previous study) | Citation | GenBank Accession numbers |
|---|---|---|---|---|---|---|---|
| Arthropoda | Insecta | Hemiptera | *Dactylopius coccus* | B | A/B | *Ramirez-Puebla et al. (2016)* | LSYX00000000, LSYY00000000 |
| | | | *Diaphorina citri* | B | B | *Subandiyah et al. (2000)*, *Lindsey et al. (2016b)* | AB038366–AB038370 |
| | | | *Gerris buenoi | B | – | – | – |
| | | | *Homalodisca vitripennis* | B | B | *Rogers & Backus (2014)* | KF636751 |
| | | | *Maconellicoccus hirsutus* | B | B | *Husnik & McCutcheon (2016)* | PRJEB12066 (European Nucleotide Archive) |
| | | | *Megacopta cribraria* | A | A | *Kikuchi & Fukatsu (2003)*, TM Jenkins, TD Eaton and C Krauss (2011, unpublished data) | AB109601, AB109602; JQ266093 |
| | | | *Mycopsylla fici* | – | – | C Fromont, M Riegler and JM Cook (2015, unpublished data) | KT273254, KT273255, KT273261, KT273277 |
| | | | *Mycopsylla proxima* | – | – | C Fromont, M Riegler and JM Cook (2015, unpublished data) | KT273257, KT273259, KT273260, KT273278 |
| | | Hymenoptera | Acromyrmex echinatior | A | A | *Frost et al. (2010)* | HM211007–HM211071 |
| | | | *Biorhiza pallida* | A | – | *Rokas et al. (2001)* | AF339629 |
| | | | *Ceratina calcarata | – | – | – | – |
| | | | Cynipini sp. | A | A | *Abe & Miura (2002)* | AB052667 |
| | | | *Diachasma alloeum | A | – | – | – |
| | | | *Diplolepis spinosa* | B | A | *Plantard et al. (1999)* | AF034987 |
| | | | *Isocolus centaureae | B | – | – | – |
| | | | *Pediaspis aceris* | A | A | *Rokas et al. (2002)* | – |
| | | | *Pseudomyrmex sp. PSW-54* | A | – | *Kautz, Rubin & Moreau (2013)* | KF015789 |
| | | | *Trichogramma pretiosum* | B | B | *Lindsey et al. (2016a)*; *Lindsey et al. (2016b)* | LKEQ00000000 |
| | | Lepidoptera | *Delias oraia | B | – | – | – |
| | | | *Operophtera brumata* | B | B | *Derks et al. (2015)* | JYPC00000000 |
| | | | *Pararge aegeria* | B | – | *Russell et al. (2012)* | KC137224 |
| | | | *Polygonia c-album* | B | B | *Kodandaramaiah et al. (2011)* | JN093149, JN093150, JN093151, JN093152, JN093153 |

**Notes.**

*Species indicated with a '*' are species that have not previously been identified, according to our literature search, to harbor *Wolbachia* strains. The supergroup classification of the *Wolbachia* strain according to this study and previously studies is listed if known.

Pascar and Chandler (2018), *PeerJ*, DOI 10.7717/peerj.5486

**Table 4  *Wolbachia* genome assemblies**  Information on *Wolbachia* draft genome assemblies. Expected host coverage is calculated as (total sequence data/host genome size). "Evidence of multiple infections" indicates whether or not the assembly contains signs pointing to multiple, distinct *Wolbachia* strains within the same biological host sample used for generating the sequence data (though some of these consisted of pooled individuals). BUSCO comp., BUSCO dup., BUSCO frag., and BUSCO missing refer to the number of BUSCO orthologs that were found to be complete and single copy, duplicated, fragmented, and missing from the *Wolbachia* assembly, out of 148 BUSCOs present in the Bacteria *odb9* reference gene set. Grey rows at the bottom of the table were omitted from the whole-genome phylogenetic analysis because the assemblies appeared less complete (as indicated by missing BUSCO genes) or showed evidence of being chimeric or a mixture of two independent strains.

| Host species/ID | Description/ common name | BioSample accession number | SRA accession numbers | Total seq. data (Gb) | Host genome size (ref.) | Expected host coverage (x) | *Wolbachia* median cov. (x) | *Wolbachia* assembly size (Mbp) | *Wolbachia* assembly N50 (kb) | BUSCO comp. | BUSCO dup. | BUSCO frag. | BUSCO missing | Evidence of multiple infections? | Sample notes |
|---|---|---|---|---|---|---|---|---|---|---|---|---|---|---|---|
| *Anopheles gambiae* | Mosquito | SAMEA3911293 | ERR1554906 ERR1554870 ERR1554834 | 9.1 | 280 Mb *Holt (2002)* | 32 | 9.0 | 1.212 | 23.12 | 125 | 0 | 2 | 21 | No | |
| *Biorhiza pallida* 1 | Wasp | SAMEA2053316 | ERR233308 | 8.7 | | | 16 | 1.249 | 9.5 | 125 | 0 | 5 | 18 | No | |
| *Biorhiza pallida* 2 | Wasp | SAMEA2053315 | ERR233309 | 8.3 | | | 16 | 1.246 | 10.29 | 128 | 0 | 3 | 17 | No | |
| *Delias oraia* | Butterfly | SAMN05712507 | SRR4341246 | 13.2 | | | 26 | 1.207 | 13.35 | 124 | 0 | 4 | 20 | No | Wild caught whole insect |
| *Diabrotica virgifera virgifera* 1 | Western corn rootworm | SAMN02373824 | SRR1106898 SRR1106897 SRR1106544 | 98.9 | 2.4 Gb (GCA_003013835.1) | 41 | 900 | 1.505 | 31 | 128 | 0 | 3 | 17 | No | 5 animals |
| *Diabrotica virgifera virgifera* 2 | Western corn rootworm | SAMN02373827 | SRR1106912 SRR1106546 | 95.1 | 2.4 Gb (GCA_003013835.1) | 40 | 750 | 1.487 | 35.01 | 128 | 0 | 3 | 17 | No | 5 animals |
| *Diabrotica virgifera virgifera* 3 | Western corn rootworm | SAMN02373842 | SRR1107707 SRR1107708 SRR1107710 SRR1107712 | 88.9 | 2.4 Gb (GCA_003013835.1) | 37 | 660 | 1.376 | 32.16 | 128 | 0 | 3 | 17 | No | 5 animals |
| *Diachasma alloeum* | Wasp | SAMN03701895 | SRR2042503 SRR2046752 | 56.8 | 390 Mb (GCA_001412515.1) | 150 | 830 | 1.377 | 21.39 | 127 | 1 | 2 | 18 | No | Adults collected from *Rhagoletis* pupae, so there may be some contamination with *Rhagoletis* DNA |
| *Diaphorina citri* 1 | Asian citrus psyllid | SAMN00100712 | SRR189238 SRR183690 | 25 | 490 Mb (GCA_000475195.1) | 51 | 180 | 1.379 | 25.7 | 128 | 0 | 2 | 18 | No | |

**Table 4** (*continued*)

| Host species/ID | Description/ common name | BioSample accession number | SRA accession numbers | Total seq. data (Gb) | Host genome size (ref.) | Expected host coverage (x) | *Wolbachia* median cov. (x) | *Wolbachia* assembly size (Mbp) | *Wolbachia* assembly N50 (kb) | BUSCO comp. | BUSCO dup. | BUSCO frag. | BUSCO missing | Evidence of multiple infections? | Sample notes |
|---|---|---|---|---|---|---|---|---|---|---|---|---|---|---|---|
| *Diaphorina citri* 2 | Asian citrus psyllid | SAMN01886038 | SRR649417 SRR649429 SRR649431 SRR649432 SRR649434 | 27.1 | 490 Mb (GCA_000475195.1) | 55 | 250 | 1.425 | 25.7 | 128 | 0 | 2 | 18 | No | |
| *Diploeciton nevermanni* | Beetle | SAMN05860871 | SRR4342174 | 24.8 | | | 63 | 1.698 | 10.59 | 124 | 0 | 5 | 19 | No | |
| *Diplolepis spinosa* 1 | Gall wasp | SAMEA3930570 | ERR1359308 | 6.8 | | | 57 | 1.398 | 12.93 | 127 | 0 | 3 | 18 | No | |
| *Diplolepis spinosa* 2 | Gall wasp | SAMEA3930574 | ERR1359312 | 7.1 | | | 53 | 1.382 | 12.13 | 121 | 0 | 3 | 24 | No | |
| *Drosophila melanogaster* 1 | Fruit fly | SAMEA3634594 | ERR1092813 ERR1092814 ERR1092815 ERR1092816 ERR1092817 ERR1092818 | 18.1 | ~175 Mb | 100 | 1600 | 1.208 | 19.19 | 127 | 0 | 3 | 18 | No | FM7a-23229-hemizygous |
| *Drosophila melanogaster* 2 | Fruit fly | SAMN04017483 | SRR2347338 | 3.5 | ~175 Mb | 20 | 21 | 1.198 | 13.83 | 125 | 0 | 4 | 19 | No | Haploid embryos; natural population |
| *Drosophila simulans* 1 | Fruit fly | SAMEA4395362 | ERR1597896 | 25.3 | ~150 Mb | 170 | 390 | 1.265 | 14.67 | 130 | 0 | 2 | 16 | No | |
| *Drosophila simulans* 2 | Fruit fly | SAMEA4394322 | ERR1597899 | 23.4 | ~150 Mb | 160 | 1100 | 1.294 | 15.57 | 130 | 0 | 2 | 16 | No | |
| *Drosophila simulans* 3 | Fruit fly | SAMEA4394323 | ERR1597900 | 37 | ~150 Mb | 250 | 1600 | 1.313 | 15.72 | 130 | 0 | 2 | 16 | No | |
| *Drosophila triauraria* 1 | Fruit fly | SAMD00051863 | DRR061000 | 22.6 | ~170 for other *Drosophila* | 130 | 150 | 1.306 | 22.94 | 129 | 0 | 3 | 16 | No | Female |
| *Drosophila yakuba* 1 | Fruit fly | SAMN04044077 | SRR2318687 | 8 | ~170 Mb | 47 | 42 | 1.254 | 14.75 | 128 | 0 | 2 | 18 | No | Pooled isofemale lines (11 lines, 4 females per line) |

**Table 4** (*continued*)

| Host species/ID | Description/ common name | BioSample accession number | SRA accession numbers | Total seq. data (Gb) | Host genome size (ref.) | Expected host coverage (x) | *Wolbachia* median cov. (x) | *Wolbachia* assembly size (Mbp) | *Wolbachia* assembly N50 (kb) | BUSCO comp. | BUSCO dup. | BUSCO frag. | BUSCO missing | Evidence of multiple infections? | Sample notes |
|---|---|---|---|---|---|---|---|---|---|---|---|---|---|---|---|
| *Drosophila yakuba* 2 | Fruit fly | SAMN04044078 | SRR2318706 | 4.4 | ~170 Mb | 26 | 35 | 1.26 | 10.24 | 127 | 0 | 2 | 19 | No | Pooled isofemale lines (11 lines, 4 females per line) |
| *Ecitophya simulans* | Rove beetle | SAMN05833357 | SRR4301374 | 22.7 | | | 170 | 1.437 | 42.92 | 127 | 1 | 3 | 17 | No | Whole insect |
| *Gerris buenoi* 1 | Water striders | SAMN02439785 | SRR1197265 | 27.4 | 990 Mb (GCA_001010745.2) | 28 | 69 | 1.538 | 13.14 | 127 | 0 | 3 | 18 | No | Female, whole individuals, adults |
| *Gerris buenoi* 2 | Water striders | SAMN02439786 | SRR1197267 | 27.8 | 990 Mb (GCA_001010745.2) | 28 | 39 | 1.537 | 13.14 | 127 | 0 | 3 | 18 | No | Male, whole individuals, adults |
| *Homalodisca vitripennis* 1 | Glassy- winged sharpshooter (leafhopper) | SAMN02209956 | SRR941995 SRR941996 SRR941997 | 107.7 | 1.45 Gb (GCA_000696855.2) | 74 | 260 | 1.675 | 14.19 | 121 | 9 | 3 | 15 | Maybe: BUSCO duplications | Lab reared Florida-strain female |
| *Maconellicoccus hirsutus* | Mealybug | SAMEA3699093 | ERR1189167 | 9.8 | 160 Mb (GCA_900064465.1) | 61 | 80 | 1.415 | 27.94 | 130 | 0 | 2 | 16 | No | |
| *Operophtera brumata* | Winter moth | SAMN03121611 | SRR1618545 SRR1618581 SRR1618582 | 22.2 | 640 Mb (GCA_001266575.1) | 35 | 28 | 1.35 | 33.34 | 129 | 0 | 1 | 18 | No | Female; adult; head and thorax; wild caught individual |
| *Pararge aegeria* | Speckled wood butterfly | SAMN02688782 | SRR1190479 | 9.8 | | | 138 | 1.282 | 83.56 | 129 | 0 | 2 | 17 | No | Whole adult, lab culture |
| *Pediaspis aceris* 1 | Gall wasp | SAMEA3925672 | ERR1355090 | 3.8 | | | 12 | 1.188 | 8.62 | 119 | 0 | 5 | 24 | No | |
| *Pediaspis aceris* 2 | Gall wasp | SAMEA3925673 | ERR1355091 | 3.8 | | | 12 | 1.174 | 8.05 | 114 | 0 | 9 | 25 | No | |
| *Polygonia c-album* | Comma butterfly | SAMN02688783 | SRR1190476 | 6 | | | 61 | 1.463 | 22.09 | 129 | 0 | 1 | 18 | No | Whole adult, lab cultured |
| *Pseudomyrmex* sp. PSW-54 | Ant | SAMN03275520 | SRR1742977 | 36.9 | 280 Mb (GCA_002006095.1 for congeners) | 130 | 195 | 1.245 | 15.91 | 127 | 0 | 3 | 18 | No | Adult female worker ant |
| *Rhagoletis pomonella* | Apple maggot fly | SAMN05388941 | SRR3900841 SRR3901027 | 23.3 | 0.97 (C-value) | 25 | 270 | 1.314 | 13.52 | 127 | 0 | 3 | 18 | No | Single adult female fly |

Pascar and Chandler (2018), *PeerJ*, DOI 10.7717/peerj.5486

**Table 4** (*continued*)

| Host species/ID | Description/ common name | BioSample accession number | SRA accession numbers | Total seq. data (Gb) | Host genome size (ref.) | Expected host coverage (x) | *Wolbachia* median cov. (x) | *Wolbachia* assembly size (Mbp) | *Wolbachia* assembly N50 (kb) | BUSCO comp. | BUSCO dup. | BUSCO frag. | BUSCO missing | Evidence of multiple infections? | Sample notes |
|---|---|---|---|---|---|---|---|---|---|---|---|---|---|---|---|
| *Trichogramma pretiosum* | Wasp | SAMN02439301 | SRR1191749 SRR1191750 SRR1191751 SRR1191752 SRR1191753 | 68.6 | 190 Mb (GCA_000599845.3) | 360 | 50 | 1.097 | 51.37 | 127 | 0 | 3 | 18 | No | |
| *Acromyrmex echinatior* | Ant | SAMEA762107 | ERR034187 ERR03416 | 13.2 | 300 Mb (GCA_000204515.1) | 44 | 56 | 1.611 | 4.66 | 104 | 0 | 8 | 36 | No | 1 male |
| *Bembidion lapponicum* | Beetle | SAMN04276907 | SRR2939026 | 8.5 | | | 11 | 1.151 | 1.79 | 70 | 0 | 13 | 65 | No | Adult, whole body |
| *Biorhiza pallida* 3 | Wasp | SAMEA2053314 | ERR233313 | 4.1 | | | 4.2 | 0.645 | 0.7 | 3 | 0 | 8 | 137 | No | |
| *Callosobruchus chinensis* | Bean weevil | SAMN02313283 | SRR949786 SRR952345 | 32.1 | 0.75 (C-value) | 44 | 340 | 2.894 | 4.06 | 78 | 6 | 12 | 52 | Yes: bimodal coverage distribution; assembly size; BUSCO duplications | Male, head, thorax, feet |
| *Ceratina calcarata* | Carpenter bee | SAMN04210145 | SRR2912519 | 16 | 200 Mb (GCA_001652005.1) | 80 | 11 | 1.053 | 1.96 | 55 | 0 | 12 | 81 | No | 1 haploid male |
| Cynipini 1 | Oak gall wasp | SAMEA1965365 | ERR233303 ERR233304 ERR233305 | 9.4 | | | 20 | 1.216 | 6.57 | 96 | 0 | 13 | 39 | No | |
| Cynipini 2 | Oak gall wasp | SAMEA2053318 | ERR233306 | 7.3 | | | 17 | 1.182 | 3.32 | 82 | 0 | 21 | 45 | No | |
| *Dactylopius coccus* | Domestic cochineal | SAMN02725055 | SRR1231828 SRR1231831 SRR1231832 | 6.2 | 21.1 Mb (estimate given in from NCBI BioSample entry) | 290 | 110 | 2.563 | 5.95 | 47 | 80 | 2 | 19 | Yes: assembly size; BUSCO duplications; bimodal coverage distribution | Bulk sample of 50 Oaxacan Mexican grana |
| *Drosophila melanogaster* 3 | Fruit fly | SAMN05417645 | SRR3931592 | 3.4 | ~175 Mb | 19 | 6.5 | 1.171 | 6.14 | 102 | 0 | 15 | 31 | No | Adult male whole body, wild caught from Africa |
| *Homalodisca vitripennis* 2 | Glassy-winged sharpshooter (leafhopper) | SAMN02209957 | SRR941998 | 39.5 | 1.45 Gb (GCA_000696855.2) | 27 | 64 | 1.803 | 17.02 | 109 | 23 | 3 | 13 | Maybe; BUSCO duplications | Lab-reared Florida-strain male |
| *Isocolus centaureae* 1 | Gall wasp | SAMEA3930555 | ERR1359249 | 3.4 | | | 7.9 | 0.998 | 2.19 | 63 | 0 | 16 | 69 | No | |

**Table 4** (*continued*)

| Host species/ID | Description/ common name | BioSample accession number | SRA accession numbers | Total seq. data (Gb) | Host genome size (ref.) | Expected host coverage (x) | *Wolbachia* median cov. (x) | *Wolbachia* assembly size (Mbp) | *Wolbachia* assembly N50 (kb) | BUSCO comp. | BUSCO dup. | BUSCO frag. | BUSCO missing | Evidence of multiple infections? | Sample notes |
|---|---|---|---|---|---|---|---|---|---|---|---|---|---|---|---|
| *Isocolus centaureae* 2 | Gall wasp | SAMEA3930556 | ERR1359250 | 3.4 | | | 7.7 | 0.965 | 2.31 | 63 | 0 | 14 | 71 | No | |
| *Megacopta cribraria* | Stink bug | SAMN02313994 | SRR1145746 | 5.7 | | | 31 | 2.097 | 1.74 | 75 | 4 | 13 | 56 | Yes: assembly size; BUSCO duplications; possible bimodal coverage distribution | |
| *Mycopsylla fici* 1 | Fig psyllid | SAMN04226368 | SRR2954433 | 0.9 | | | 9.9 | 1.171 | 2.32 | 76 | 0 | 6 | 66 | No | |
| *Mycopsylla fici* 2 | Fig psyllid | SAMN04226369 | SRR2954467 | 0.8 | | | 12 | 1.238 | 2.68 | 77 | 0 | 14 | 57 | No | |
| *Mycopsylla proxima* | Psyllid | SAMN04226370 | SRR2954473 | 1.1 | | | 5.7 | 0.364 | 0.76 | 0 | 0 | 3 | 145 | No | |
| *Rhagoletis zephyria* | Tephritid fly | SAMN04977950 | SRR3670118 SRR3670117 SRR3670120 | 132.9 | 1.1 Gb (GCA_001687245.1) | 120 | 1200 | 1.881 | 11.19 | 80 | 51 | 2 | 15 | Yes: bimodal coverage distribution; assembly size; BUSCO duplications | Single adult female fly |

our purposes this should be a reasonable approximation. We then estimated the expected sequencing depth of the host by dividing the total amount of sequencing data by the estimated host genome size. Although genome size data on some host species was lacking, large differences in sequencing depth between the host and endosymbiont support active *Wolbachia* infections in several species, including *Anopheles gambiae*, *Diabrotica virgifera*, *Diachasma alloeum*, *Diaphorina citri*, several *Drosophila* species, *Homalodisca vitripennis*, *Rhagoletis pomonella* and *R. zephyria*, *Trichogramma pretiosum*, *Callosobruchus chinensis*, *Ceratina calcarata*, and *Dactylopius coccus* (Table 4).

In a few cases, there was evidence of multiple infections in a single sample. This evidence included an unusual number of duplicated BUSCO reference genes in the assembly (e.g., *Homalodisca vitripennis* 1), the presence of multiple peaks in the coverage distribution histogram (e.g., *Callosobruchus chinensis*), assembly sizes much larger than previously sequenced Wolbachia genomes (e.g., *Dactylopius coccus*), or some combination of these (Table 4).

### *Wolbachia* phylogeny

All phylogenetic trees based on individual or concatenated datasets using the *ftsZ* and *groE* sequences show two distinct branches representing supergroups A and B (Fig. 1; Figs. S1–S2). The tree resulting from the concatenated dataset has the most robust bootstrap support for most clades. Positive control samples that were included in the phylogeny cluster with other control samples of the same known supergroup. Of the species where *Wolbachia* had been previously unidentified, according to all trees, the strains isolated from *D. alloeum* falls within supergroup A, while the *B. lapponicum, I. centaureae, G. buenoi, D. nevermanni* and *D. oraia* isolates all fall within supergroup B (Figs. 1 and 2, Figs. S1–S2).

The phylogeny generated from whole-genome sequencing data (Fig. 2) was similar in overall topology to the trees based on *ftsZ* and *groE*, with two clear clades representing supergroups A and B, but with higher bootstrap support for most branches.

## DISCUSSION

### Observed low infection rates

While *Wolbachia* is estimated to infect between 20–76% of arthropod species (*Werren, Windsor & Guo, 1995*; *Jeyaprakash & Hoy, 2000*), in this set of data only 11.8% of species tested positive. Given the source, this low rate of infection can be hypothesized to be the result of five possible scenarios: (1) Underrepresentation in the amount of data available per host species. For example, only 43 out of the 288 (14.9%) species and subspecies tested had ≥10 samples available in the SRA that met the criteria of this study (Table S2). When >100 individuals are tested for *Wolbachia*, results are skewed towards finding a positive sample (*Hilgenboecker et al., 2008*). (2) Bias in the source of the samples. Sources vary between wild-caught individuals, lab stocks, and unreported sources. Since the phenotypic consequences of *Wolbachia* are well established, if uninfected individuals are needed for a study they may be selectively chosen (see *Đorđević et al., 2017*; *Becking et al., 2017*), or the researchers may even actively treat infections with antibiotics (*Dobson & Rattanadechakul, 2001*; *Casiraghi et al., 2002*; *Koukou et al., 2006*) or increased rearing temperature. In those

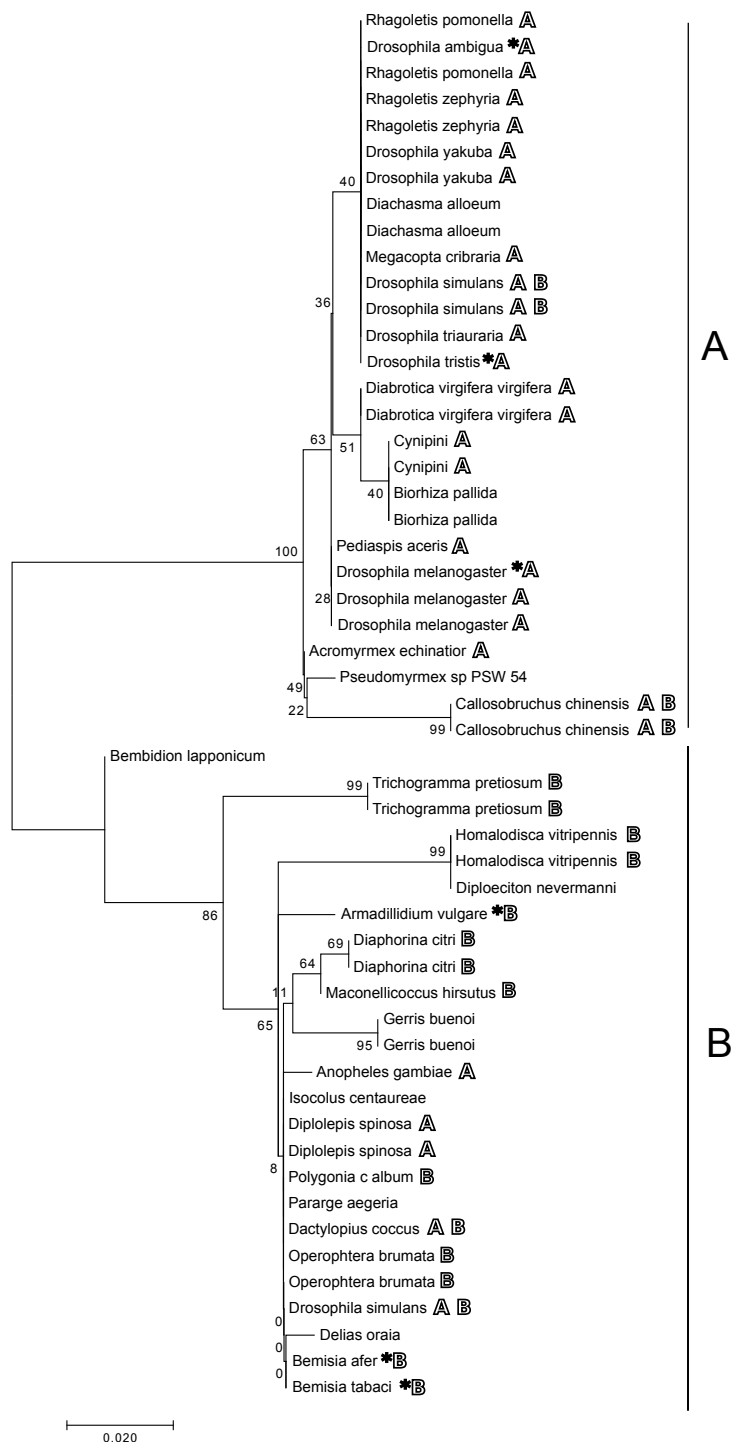

**Figure 1 Concatenated phylogeny.** Molecular phylogenetic analysis by maximum likelihood based on the concatenated dataset containing *ftsZ* and the *groE* operon (total of 1,381 nucleotide positions). Bold letters next to host species names indicate supergroup relationships of *Wolbachia* isolates identified in previous studies for each host. Asterisks indicate reference sequences (see also Table 2).

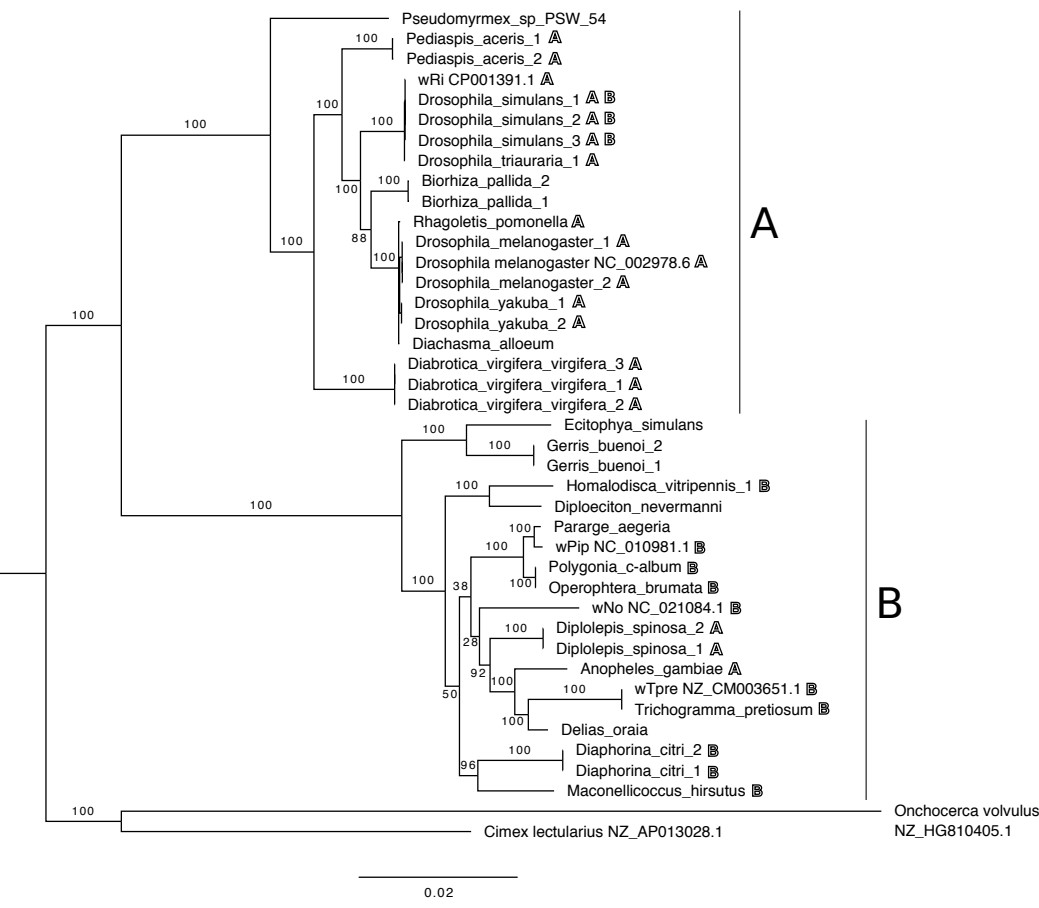

**Figure 2  Whole-genome phylogeny.** Maximum likelihood phylogeny based on whole-genome sequence data of *Wolbachia* isolates assembled here and previously sequenced reference *Wolbachia* genomes (indicated by samples with associated accession numbers), with a total of 133,744 nucleotide positions. Numbers by nodes indicate bootstrap support based on 200 replicates. Bold letters next to host species names indicate supergroup relationships of *Wolbachia* isolates identified in previous studies for each host. Isolates with accession numbers listed represent reference genome sequences from other studies.

cases the sequencing data will consequently test negative for *Wolbachia* using the methods employed here. (3) Tissue sampled. In some species infection has only been detectable in the gonads, indicating that infection density in somatic tissue may be variable or low (*Dobson et al., 1999*). For many samples in the SRA, specific tissue has not been indicated. (4) Bioinformatic removal of bacterial contaminants. Even if *Wolbachia* is sequenced with the host's DNA, the researcher may have eliminated these reads bioinformatically before depositing the reads as relatively standard practice in sequence processing (*Kunin et al., 2008*; *Schmieder & Edwards, 2011*; *Derks et al., 2015*). (5) False negatives. It is possible that some infections may have been missed due to the limited set of available reference genes; more divergent strains might not have been detected in these analyses.

## Strain supergroup affiliation

For 13 of the species that tested positive, previous information was available about supergroup affiliations of *Wolbachia* strains that have been found to infect them (Table 3). Our results are mostly consistent with previously reported phylogenetic relationships. Previously, *C. chinensis, D. coccus,* and *D. simulans* have been found to be infected with A and/or B strains (*Kondo, Shimada & Fukatsu, 1999*; *Kondo et al., 2002*; *Riegler et al., 2004*; *Ellegaard et al., 2013*; *Ramirez-Puebla et al., 2016*). Here, evidence of both A and B supergroup strains was found in *D. simulans* (Fig. 1), though the whole-genome phylogeny was somewhat inconsistent here, suggesting possible recombination for some genes. Moreover, while the single-gene phylogenies suggested that the endosymbionts of these *C. chinensis* and *D. coccus* samples were members of supergroups A and B, respectively, the whole-genome assemblies for both of these endosymbionts contained strong evidence of dual infections, so we cannot rule out the presence of both A and B supergroup strains in these samples.

  *Wolbachia* infection has also been documented prior to this study in *B. pallida, P. aegeria*, and *P. sp. PSW-54* but the supergroup relationships were not reported (*Subandiyah et al., 2000*; *Rokas et al., 2001*; *Russell et al., 2012*; *Kautz, Rubin & Moreau, 2013*; *Baldini et al., 2014*). Our concatenated results suggest that a supergroup A strain infects *B. pallida* and *P. sp. PSW-54*, while a B strain infects *P. aegeria*.

  Two species showed a different supergroup strain than what has been previously reported—*D. spinosa* and *A. gambiae*. *D. spinosa* has previously been identified to harbor a supergroup A strain, but here we discovered an infection that clusters within supergroup B. It may be possible for *D. spinosa* to harbor both A and B strains since other species in the genus have been shown to have supergroup B infections (*Plantard et al., 1999*).

  Particularly notable is our identification of a supergroup B strain in *A. gambiae.* Anopheline mosquitoes were once thought to lack infection by *Wolbachia* in nature (*Kittayapong et al., 2000*; *Ricci et al., 2002*; *Rasgon & Scott, 2004*), though they are capable of experimental infection in the lab (*Hughes et al., 2011*). However, there have been recent reports of natural infections in wild populations (*Baldini et al., 2014*; *Gomes et al., 2017*). In particular, a supergroup A strain was found to infect *A. gambiae* mosquitoes in Mali that reduces the transmission of the malaria parasite (*Gomes et al., 2017*). The strain identified here clearly belongs to supergroup B, and is related to strains infecting Hymenoptera and Lepidoptera, rather than fleas (Siphonaptera) like the previously identified supergroup A strain. In addition, other recent surveys have found evidence of diverse *Wolbachia* strains, including supergroup B strains, within the *A. gambiae* species complex (*Ayala et al., 2018*; *Jeffries et al., 2018*). Combined, these results suggest that the diversity of *Wolbachia* infections in *Anopheles* may be currently underappreciated. Importantly, we have good evidence that the sample here is an actual *Wolbachia* infection rather than an integrated piece of *Wolbachia* DNA in the host genome. First, we assembled a nearly complete *Wolbachia* genome from this dataset; more importantly, given that the *A. gambiae* genome is roughly 280 Mb (*Holt, 2002*), and this dataset contained roughly 9.1 Gb of raw sequence reads, we would expect to have roughly ~32× coverage of the host genome, but the

*Wolbachia* genome had only ∼9×coverage, suggesting that *Wolbachia* DNA was present at lower densities in this sample than the host DNA.

The assembled *ftsZ* and *groE* sequences from *C. calcarata, E. simulans. M. fici,* and *M. proxima* assemblies were too short to be included in our individual gene-based phylogenetic reconstruction; the infection density, and thus the sequencing coverage, for these species may have been too low to yield reliable assemblies for these genes. We were able to assign the *E. simulans* infection to supergroup B based on its draft genome sequence, but the supergroup relationships for the others are still unknown. Previously, *Wolbachia* sequence information has been isolated in *M. fici,* and *M. proxima* (Fromont et al., unpublished data; Table 3) but supergroup affiliation was not suggested. According to our literature search this is the first detection of *Wolbachia* in *C. calcarata* and *E. simulans.* The other six previously unidentified species infections were included in the phylogeny. *D. alloeum* clustered with known supergroup A infections while *B. lapponicum, D. oraia, D. nevermanni, G. buenoi, I. centaureae* isolate clustered within the supergroup B clade.

Finally, our phylogeny also offers some hints into possible mechanisms of horizontal transmission of *Wolbachia* infections. In particular, the strain identified here infecting *Diachasma alloeum* is closely related to the strain found in *Rhagoletis pomonella* (as well as *D. melanogaster* and *D. yakuba*). This is intriguing because *D. alloeum* is a parasitoid wasp that uses *R. pomonella* and *R. mendax* as its host (*Maier, 1981*; *Stelinski, Pelz & Liburd, 2004*), suggesting that this may represent a natural horizontal transfer of *Wolbachia* from one lineage to another; previous studies have found evidence of horizontal transmission between predators and prey or hosts and parasites (*Heath et al., 1999*; *Le Clec'h et al., 2013*). However, contamination by host material in parasitoid samples, or vice versa, could also explain this outcome, so this result should be interpreted cautiously until this path of transmission can be experimentally confirmed.

### Multiple infections and integration of wolbachia into the host genome

Double (*Perrot-Minnot, Guo & Werren, 1996*; *Narita, Nomura & Kageyama, 2007*) and even triple *Wolbachia* infections (*Rousset, Braig & O'Neill, 1999*; *Kondo et al., 2002*) have been reported in arthropod populations and individuals, both naturally and through experimental injection. The initial screening methods presented here are not capable of identifying multiple infections because we only looked for a positive or negative test result and then used only the single longest contig for phylogenetic construction. In conventional PCR there is a tradeoff between specificity and sensitivity of primers; additionally no one primer is capable of identifying *Wolbachia* in all samples (*Simões et al., 2011*). PCR is useful in initial infection confirmation but sequencing is usually necessary to confirm group relationships. Techniques used to identify multiple infections currently include quantitative PCR with highly specific primers (*Kondo et al., 2002*; *Narita, Nomura & Kageyama, 2007*), cloning and sequencing (*Jamnongluk et al., 2002*), and Southern hybridization (*Perrot-Minnot, Guo & Werren, 1996*).

We were able to identify evidence of possible multiple infections through genome assembly. In some cases, the assembly was approximately double the expected size, contained a large number of duplicated genes, or showed evidence of multiple peaks in a

coverage histogram, all of which are signs of infection by multiple, independent strains. Again, these results should be interpreted cautiously pending experimental validation. For instance, some of the multiply infected samples consisted of pooled DNA from multiple individuals (e.g., *Drosophila yakuba* and *Diabrotica vinifera*), so the "multiple" infection might simply result from different individuals in the sample harboring different endosymbiont strains. Nevertheless, these results show that high-throughput sequencing can be a powerful way to detect multiple infections, especially when a priori sequence information for designing strain-specific primers is unavailable.

A related issue is that *Wolbachia* DNA is frequently integrated into host genomes (*Vavre et al., 1999*; *Leclercq et al., 2016*); in some cases, these insertions even consist of nearly whole *Wolbachia* genome sequences (*Dunning Hotopp et al., 2007*). This complicates our analyses because some of the identified "infections" could actually be *Wolbachia* DNA integrated into the host genome; in fact, horizontally transferred Wolbachia DNA has already been identified in four orders which are all represented by the positive results in this study, Coleoptera, Diptera, Hemiptera, and Hymenoptera  (*Hotopp, 2011*). We were able to rule out horizontally transferred DNA in some, but not all, cases of positive samples, using sequencing depth information; if the sequencing depth of the assembled *Wolbachia* contigs differs from the sequencing depth of the host's nuclear DNA, that suggests a true, active infection. True infections could also be validated experimentally when necessary, for example, using fluorescence in situ hybridization (*Hughes et al., 2011*). Either way, horizontally transferred *Wolbachia* DNA would still indicate that a species at least had a history of infection at some point in the past.

This work shows that it is often possible to assemble draft genomes of endosymbionts from host DNA, similar to previous studies in which *Wolbachia* genomes were assembled from sequencing host organisms (*Ghedin et al., 2004*; *Salzberg et al., 2005*; *Richardson et al., 2012*; *Saha et al., 2012*; *Campana, Robles García & Tuross, 2015*; *Derks et al., 2015*; *Lindsey et al., 2016b*), even when the endosymbiont was not the focus or original reason for performing the sequencing in the first place. Although they may be fragmented, these draft genomes can still provide valuable information about the phylogenetics and evolution of the endosymbiont. While *Wolbachia* is relatively well studied, there are many other endosymbionts that have received less attention, such as some *Spiroplasma, Cardinium, Arsenophonus,* and *Flavobacetrium* species (*Duron et al., 2008*), and others await discovery. This study shows that extensive field sampling may not even be necessary to get a better understanding of the diversity of these endosymbionts; the sequencing data are probably already available in public databases. With the right reference databases and metagenomics software, there is a lot of potential to learn more about these endosymbionts just from already existing resources.

## CONCLUSIONS AND RECOMMENDATIONS

*Wolbachia* is a well-known endosymbiont of many arthropod species and while standard *Wolbachia* diagnostic techniques utilize various *Wolbachia* primers to confirm infection via PCR (*Simões et al., 2011*) there are trade-offs that limit large scale surveys. Here, we

present a method to identify *Wolbachia* bioinformatically using publicly accessible host raw sequencing data. In eight arthropod species, *Wolbachia* was identified where infection has not previously been reported, and in 27 other arthropod species infection was confirmed. Isolates of *Wolbachia* from positive samples all clustered within either supergroups A or B, and for seven of the newly identified hosts we identified the supergroup of the strain. From these isolates we assembled draft *Wolbachia* genomes, which provided robustly supported phylogenetic information as well as information about potential HGT events or signs of multiple infection.

These results highlight the importance of depositing raw sequencing datasets to public archives like the NCBI SRA and the value that they have in studying endosymbionts. At the same time, we offer some suggestions for best practices when depositing sequence data into public archives to maximize its usefulness for future researchers (*Wilkinson et al., 2016*; *Griffin et al., 2017*). First, we encourage everyone performing high-throughput sequencing to deposit their data into public databases like the NCBI SRA, where it can easily be searched and accessed, as opposed to depositing only in smaller, taxon-specific databases or personal/lab web sites. Second, data should be minimally filtered; while ''contaminant'' sequences like endosymbiont DNA may be a nuisance to those who generated the data, they may be of interest to others. Finally, all sequence data should be accompanied by as much metadata as possible. Without this information, interpreting results can be difficult. For example, many of the sequences we used in this study lacked detailed information about the source of the DNA in the associated BioSample entries (e.g., whether it came from a lab strain or wild-caught specimens, its geographic origin if field collected, whether it was from a single individual or a pooled sample, whether the specimen was male or female, whether it was a whole body or specific tissues, etc.). Including this information would have helped us better understand possible biases in the dataset, such as how well the results may reflect the frequency of infection in natural populations, or whether a sample might give a false negative result because *Wolbachia* is not present at high densities in the tissues sampled for DNA.

## ACKNOWLEDGEMENTS

We thank P Brannock, S Borrelli, and the editor and three anonymous reviewers for their helpful feedback and suggestions on earlier versions this manuscript. We also thank V Buonaccorsi, C Walls, and GCAT-SEEKquence for computing support, as well as the National Center for Genome Analysis Support at Indiana University.

### Funding

This work was supported by the National Science Foundation (NSF DEB-1453298 to Christopher Chandler, and ABI-1458641 to Indiana University). The funders had no role in study design, data collection and analysis, decision to publish, or preparation of the manuscript.

### Grant Disclosures

The following grant information was disclosed by the authors:
National Science Foundation: NSF DEB-1453298.
Indiana University: ABI-1458641.

### Competing Interests

The authors declare there are no competing interests.

### Author Contributions

- Jane Pascar and Christopher H. Chandler conceived and designed the experiments, performed the experiments, analyzed the data, contributed reagents/materials/analysis tools, prepared figures and/or tables, authored or reviewed drafts of the paper, approved the final draft.

### Data Availability

The raw data and code are provided in the Supplemental File.

### Supplemental Information

Supplemental information for this article can be found online at http://dx.doi.org/10.7717/peerj.5486#supplemental-information.

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
