# Peer review of "A bioinformatics approach to identifying Wolbachia infections in arthropods"

_PeerJ, doi:10.7717/peerj.5486_

## Round 0.1 · original submission · Major Revisions

Dear Jane and Christopher:

We have received three reviews of your work, and as you will see, two reviewers have made many suggestions that have great potential to improve your work. Should you decide to submit a revision, please take these suggestions seriously. In particular, please pay careful attention to the concerns regarding certain genes and known recombination across different Wolbachia strains and species. Also, please provide more information RE methods (e.g., alignment parameters, exclusion criteria, mapping/assembly parameters, etc.). There are also numerous suggestions on how to improve the writing and organization of the manuscript. Overall, the careful attention provided by the reviewers will undoubtedly improve this work, and I look forward to receiving your revision should choose to continue forward with publishing in PeerJ. Best of luck,

-joe

Reviewer 1 ·

Basic reporting

Thank you for the opportunity to review the manuscript submission “A bioinformatics approach to identifying Wolbachia infections in arthropods”. Pascar and Chandler present a method by which to screen SRA libraries from arthropod sequencing projects for evidence of Wolbachia infection. Wolbachia is the most common symbiotic bacterium across insects, and has sculpted the evolutionary trajectories of many lineages, allowing for niche expansion and creating reproductive barriers through various mechanisms. This is a fun study that is a good match for PeerJ, and takes advantage of the vastly growing amount of short read data with the intent of picking up previously undescribed associations with Wolbachia. The manuscript is nicely written and extensively referenced. I was happy to see the scripts and lots of intermediate data in the supplemental. While there are some interesting findings (ex., newly described Wolbachia associations, and supergroup assignments for previously un-assigned strains), some of the methods are problematic and should be re-visited.

Experimental design

Major Comments:

These are ones that would affect the phylogenetic reconstructions, and prevalence estimations.

1) The choice of reference loci (ie., wsp, ftsZ, groE) should be fine for detecting Wolbachia reads, but the use of wsp in phylogenetic reconstruction is problematic due to extensive recombination. The authors touch on this with the now retired Supergroup G, and reference Baldo and Werren (2007) (line 62), which is a great example of the problematic nature of wsp.

2) It looks like there are mate-pair libraries included in the samples. Are different mapping/assembly parameters used for MP vs PE read pairs to account for different directionalities? Along those lines, were different fragment sizes controlled for? You may be missing some hits if the pairs are not handled correctly.

3) Most of the host species have multiple SRA libraries that originated from the same BioProject/DNA extraction. Wolbachia sequences were assembled separately for each of these positive libraries, and then these “multiple strains” of Wolbachia were used in phylogenetic reconstruction. However, these data represent the same Wolbachia strain, just a different library preparation. This is misleading, as it looks like the authors have recovered multiple strains from the same species.

4) I would imagine that the size of the host genome would be an important consideration for the number of reads used for Wolbachia detection, as the ratio of host:Wolbachia DNA will be different and have an effect on detection.


Secondary Comments:

There are a few things that, while they should not have an effect on the major findings of the paper, could benefit from some smoothing out.

5) The authors choose to search arthropod SRA libraries, but blast them against reference loci from supergroups A-D (with C and D infecting nematodes). An explanation of why they included C and D loci would be beneficial, as we would not expect to recover these supergroups from arthropod samples.

6) How/why reference gene alleles were chosen could benefit from some discussion. For example, there are multiple wsp alleles from Nasonia vitripennis, unequal numbers of alleles chosen per reference gene, some of the groE operon alleles are not complete, etc. I really like the way Table 2 is laid out and the choice of “positive controls” in the analysis. Why not use these positive phylogenetic controls as the reference loci?

7) I’m having difficulty matching what is in the scripts with what is in the methods. In the scripts it looks like first the libraries were pre-screened for wsp hits, and then positives were reanalyzed to pull down the other loci?

8) Is the non-coding region of the groE operon used in phylogenetic reconstruction?

9) Line 144: What length of matching basepairs was considered “sufficient”?

10) Line 145: What parameters were used for Gblocks?

Validity of the findings

11) The bootstrap values seem very very low (0’s, for example) for some clades. I would re-visit these.

12) It would be nice to see some statistics on read coverage that the authors obtained for their reference loci.

13) I like the discussion of the factors that can contribute to false negatives/positives and skewed Wolbachia prevalence using the presented method. While it would take significant additional work, the authors could include host loci to see if Wolbachia and host have equivalent or drastically different read coverage (to get at HGT vs infection). Similarly, the authors could look at their libraries and see if those with positive hits were more likely to be “wild-caught” (addressing hypothesis #2 in the discussion).

Additional comments

Minor Comments:

14) The last sentence of the abstract is unclear: what is meant by “similar methods can be replicated”?

15) The introduction is generally well written though I might suggest reordering such that lines 51-56 go along with the first paragraph in which the authors discuss Wolbachia prevalence.

16) Line 92: Insecta is a class, not an order.

17) Throughout the methods “samples” is used a lot, which I think is confusing given my earlier comment (#3). I would suggest using, “species”, “samples”, and “libraries”, or something similar to help differentiate between these levels of organization.

18) Line 130: s/Velveth/Velvet/

19) Table 1 would benefit from some reformatting. The definition lines (taken from NCBI) are not descriptive – ex., the first 5 ftsZ alleles have no host species listed. My preference would be to show host species, supergroup (if known), length of the sequence, and citation, similar to what you have done for table 2, which I like very much.

20) The phylogenetics methods should be re-located from the figure captions to the methods section.

21) Table 3: what is the difference between “?“ and “-”?

22) The literature search in the supplemental can be included in the main manuscript text.


Summary:

There are some kinks that need to be worked out in regards to the methods, but I think the take away of this manuscript is a nice contribution and I hope to see an improved version.

Reviewer 2 ·

Basic reporting

See below

Experimental design

See below

Validity of the findings

See below

Additional comments

This manuscript by Pascar & Chandler presents the results of a Wolbachia screen within short read samples stored in the SRA public database of the NCBI. The screen was performed by (partly) downloading 2,545 short read libraries generated from 288 species and mapping these to three Wolbachia genes. The matching reads were assembled and their phylogeny analysed together with reference sequences to determine supergroup affiliation of the detected strains. This way, 8 novel Wolbachia – host associations could be determined.
I think the idea of this study is very cool and the analyses seem to be well planned and performed. Short read libraries are indeed a valuable source for symbiont data that is often neglected, and this can’t be stressed enough – this paper is a welcome contribution in this respect.

I have however two major reservations that I feel should be addressed before this should be published, one regarding analytical approach, and one regarding the form:

1) The data seem to be of mostly preliminary nature, i.e., it appears that the authors have stopped at a point where things become most interesting. The approach, while not totally new, is certainly well suited to detect bacterial symbionts in arthropods, so why not check for more (all?) endosymbionts known from arthropods? As opposed to PCR screens, this requires little effort and would hardly take more time than checking for Wolbachia only. Furthermore, 288 species does seem like a low number for all arthropod species in the SRA database (maybe I am wrong there). I fear that the authors’ rather restrictive filtering has led to the exclusion of a number of potentially interesting samples. I think there is a really good chance of some cool findings with such a screen. There have been countless PCR based screens and meta-analyses investigating the incidence of various endosymbionts. Here, you present an alternative, more or less unbiased dataset, potentially providing independent evidence for these previous analyses.
As an alternative a more in-depth screen, why not try to extract whole Wolbachia genomes from at least some of the detected associations? This would demonstrate it actually is possible to ‘assemble complete Wolbachia genomes’ as the authors mention in the conclusions section, and I know that the Wolbachia community would welcome such an effort.
I have other comments about the methodology (listed below), but I think this is my biggest issue with the manuscript.

2) Related to point 1), I think the authors should clarify the narrative of the manuscript. In its current form, the reporting switches between a ‘proof of concept’ kind of paper and a Wolbachia screen with a new approach. I feel that the latter is the better option given that the concept is not entirely new. However, as mentioned in point 1), this screen should be extended to potentially yield more meaningful insights.

Please find more details on these points and other, minor comments below. I hope you will find these constructive:

Introduction
The introductory paragraphs are mostly about the biology of Wolbachia. There’s nothing wrong there, but I think this could be condensed and the readers referred to one of the excellent Wolbachia reviews around. Instead, the authors should expand more on the motivation behind their study and background of their methodology. They should also mention other studies with a similar scope (they do this later in the discussion section) and how their study differs from these previous studies. A number of bioinformatic tools exist that enable the extraction of microbial reads from eukaryotic short reads (e.g., Blobtools or Anvi’o), maybe these should also be mentioned. You may also want to have a look at the recently published study on honey bee symbionts that essentially uses the same approach as your study (https://peerj.com/articles/3529/).

L34
Change ‘α-proteobacteria’ to ‘α-Proteobacterium’

L36
Suggest to change ‘distribution’ to ‘incidence’ or ‘frequency’

L72–73
ftsZ and groEL don’t exactly qualify as fast evolving.

L77–79
wsp is highly recombinant and therefore inherently unsuited as phylogenetic marker or to ‘identify groups and strains’ of Wolbachia

L106ff
As mentioned above, it is unclear why you excluded other sequencing technologies or single library layouts. You may be missing quite a few interesting samples.

L122
What was the rationale behind choosing these particular loci? When using multiple loci, it would make sense to use the MLST genes, this would enable comparison with many other strains. Much more important than multiple different genes (why should one not be sufficient?) however, is the inclusion of more divergent Wolbachia lineages. You mention many Wolbachia supergroups in the introduction, and for all of these, there is some genetic data somewhere that can be used for screening. Of the four supergroups you do employ, two (C & D) were never found in arthropods, so you basically only searched with A and B groups. I suspect that may also be a reason why these were the only hits that were recovered. Further, as mentioned above, I think it would be worthwhile to check for other common symbionts (Rickettsia, Spiroplasma, Arsenophonus, Cardinium, Sodalis, etc).

L123
Does this mean 98bp when summing the length of all reads? Otherwise you may have excluded short reads from earlier Illumina platforms (50bp or 75bp reads were not uncommon).

L127
Were these randomly chosen? It may be better to assemble all the samples and then chose the best contigs. Assembling these tiny regions shouldn’t be computationally prohibitive. Also, it might lead to the detection of multiple different strains in a single host species.

L144
What did you consider of ‘sufficient length’?

Results section
It would be nice to have a graphical overview of the taxonomy of the samples, i.e, how many samples were from each of the investigated taxonomic groups. Also, was there any evidence for geographic clustering? Did you check for tissues from which the samples have originated? I think that looking at the metadata might reveal some interesting patterns.

L153
For the positive samples, were always all of the genes recovered?

L167
I think that four tree figures are not really needed, as they all show similar information. Also, differences between the trees are not really discussed anywhere. I suggest to keep the concatenated tree in the main manuscript and move the rest into the supplement.

L193
I would disagree that the threshold used (three matching reads) is to strict. The low number of infected species might be due to the reference sequences used.

L218
One of the most interesting findings of this paper is hidden here: the detection of supergroup B Wolbachia in Anopheles gambiae. Wolbachia seems to be mostly absent in Anopheles, and it has been shown that there an incompatibility between its native microbiome and Wolbachia may be the reason for this (http://www.pnas.org/content/111/34/12498.short). The only study (I know of) claiming to have found infections in natural populations is the one you cite (Baldini et al 2014). Frankly, this paper is not very convincing – they sequenced two complete Illumina lanes and only got a handful of Wolbachia reads which they claim belongs to a distinct phylogenetic lineage (based on questionable methods in my opinion).
If you can demonstrate that this is indeed a supergroup B Wolbachia strain, this would be very neat and interesting to many people, both working on Anopheles and Wolbachia. Even better, if you can assemble the genome of this Wolbachia strain (which should be feasible), and show that this is not only a chromosomal insertion of the host, this might make for another very interesting paper!

L239
HGT could be assessed by checking read pairs after mapping. If only a single read of a pair has mapped to the Wolbachia target, it can be tested if the other read matches to a host chromosome.

Final comment
This may seem like a lot of nagging, and I apologize for this. However, this is because I really feel that with a little effort, this paper has the potential to become an exciting contribution.

·

Basic reporting

This is a very well put together paper. The structure, language and context provided are more than appropriate for publication. I would also like to commend the authors for the completeness of the submission, for not only including raw data, but the code and literature search methodology to reproduce the work.

Experimental design

I believe the experimental design element meets the criteria for publication. Methods are clear, rigorous and clearly address the stated research goals and I only have one minor comment here.

Minor comment:
Diagnosing Wolbachia infection [lines 119-125]. My reading of this suggests that +ve infection was declared even if only one of the reference genes met the matching reads criteria - is this correct? If so, how robust do you think this criteria is - did you do any benchmarking with known infected samples? Also, how do your infection rates change when you set the threshold to be that i) at least 2 and ii) all reference genes must meet the matching criteria?

Validity of the findings

I feel the data is sound and the conclusions are well stated.

---

## Round 0.2 · Minor Revisions

Dear Drs. Pascar and Chandler:

Thanks for revising your manuscript based on the minor concerns raised by the reviewers. I have now received feedback from the three original reviewers of your work, and as you will see, while very favorable, a little more needs to be done. The reviewers raised some relatively minor concerns about the research, and areas where the manuscript can be improved. Please consider missing references, as well as issues related to phylogeny estimation and read mapping. I agree with the reviewers, and thus feel that their concerns should be adequately addressed before moving forward.

Therefore, I am recommending that you revise your manuscript accordingly, taking into account all of the issues raised by the reviewers. I do believe that your manuscript will be ready for publication once these issues are addressed.

Good luck with your revision,

-joe

Reviewer 1 ·

Basic reporting

I am so pleased to see such a thorough set of revisions on the manuscript submitted by Pascar and Chandler. Please find attached my suggestions for some last smoothing over. I have also included some additional experimental suggestions. I do not find them necessary for publication of your results, but they might be nice quick additions, especially if you are familiar with the bioinformatics. There are a few references that could benefit from updates (please see general comments). I think this paper will be a nice addition to the literature.

Experimental design

- I maintain that wsp should not be used for phylogenetic reconstruction. I think at minimum there should be a discussion of its utility (or lack there of). The references in lines 65-67 came before the discovery of wsp recombination. I think it is fine to use wsp for detection, but why not just use ftsZ and the GroEL operon for phylogenetics?

- It isn’t clear what constitutes “unrealistic” sizes of genomes for determining HGT. Insect genomes vary in size by orders of magnitude (from <100Mbp to >6Gbs). Additionally, I do not think the coverage statistics are calculated or reported in a clear way: "Host genome size if same cov. as Wo" is not intuitive, especially when you are comparing that value in Mbp to host genome size as a C-value. The authors mention the difficulties of obtaining whole genome coverage, but maybe coverage of a single copy gene, or the arthropod BUSCO set would be helpful. Many genome sizes are missing from Table 4, but are listed on NCBI for the host genome assemblies that came from these reads (ex., Gerris buenoi = 994Mbp, Diabrotica virgifera = 2.4 Gbp, Diachasma alloeum = 388 Mbp, D. citri = 485 Mpb, and many others).

- It appears that BisSeq libraries were used in screening. These probably are not useful, as they are not the true genomic sequence, but will contain converted nucleotides from the bisulfite processing.

Validity of the findings

No issues here. Please see my more general comments below.

Additional comments

- Line 150: For the readers' convenience, please include the strain names of these genomes in the text (ex., wPip_Pel, wMel, wNo, etc).

- I think in figures 1 and 2, there are many species that should have the squares indicating supergroup was previously known (ex., For D. simulans there are many papers; many works by Stouthamer show Trichogramma pretiosum has supergroup B, including Lindsey et al 2016 Genes|Genomes|Genetics for that exact genome + placement of others like wDi from D. citri; O. brumata and D. coccus have also been previously assigned a group I believe). Many of these appear labeled in Table 3, so it isn’t clear why they are not all labeled on the figure.

- The reference for the Trichogramma pretiosum wTpre Wolbachia genome (Table 3) should be Lindsey et al 2016 Genes|Genomes|Genetics. (Currently the only reference in the citations is for the Systematic and Applied Micro paper, which talks about taxonomy and is referenced at line 48.)

- Several of your assembled Wolbachia genomes already have published assemblies from these reads/biosamples (ex., wDacA, wDacB, wTpre, etc). The wTpre genome, as well as the wDacA and wDacB genomes, were computationally separated from their respective hosts and used many of the same methods as you (ex., sequencing coverage), so may be worth referencing at lines 86 and 370. Additionally, you could align your versions with the published versions to see how well your iterative assembly method worked (Ex., use nucmer/mummer to get an estimate of the similarity).

- I like that the authors attempted to look for Wolbachia co-infections. Some of these have been published (ex., D. coccus: Ramírez-Puebla et al 2016 Genes|Genomes|Genetics), which the authors reference (line 280). It gets a little complicated, because the phylogenetics place just the B strain in D. coccus, but the Wolbachia assembly appears to be a chimera of the A and B, which is what I would expect given what we know about this dataset and the assembly method. Addressing this in the discussion would be appreciated.

- I like that the authors have urged others to include better metadata in their biosample submissions. I am not sure what the appropriate amount of filtering to suggest would be. While clearly the symbiont reads are useful, someone who is not looking for those could be lead astray by their presence. Definitely the information on filtering should be included in the metadata, and perhaps there should be options to submit both raw and filtered reads.

-What is the difference between the white and grey rows in table 4?

- Line 73-74: Stouthamer 1990 PNAS is one of the first references for this.

- Lastly, I am happy that the authors have provided the draft assemblies. I would like to state for the record, that I think they are premature for NCBI, and as such I agree with the authors’ decision to only provide them in supplement. There are other examples of Wolbachia genomes on NCBI that are chimeras and/or HGTs (ex., wAna). Unfortunately these genomes are often used in analyses and it confuses the literature. Hopefully you or others can follow up on the drafts you generated to verify their true infection status and assembly quality.

Reviewer 2 ·

Basic reporting

see below

Experimental design

see below

Validity of the findings

see below

Additional comments

I am very happy to see the revised work by Pascar & Chandler much improved, most importantly by the assembly of the Wolbachia genomes. This was a big effort, but I think it was worth it. Also, the narrative has been tightened and the text is much better to follow than the original version. I want to congratulate the authors for this contribution. I have in general no objections against this being published, but found one important issue that needs fixing.

From the scripts you submitted as Supplementary material (which I find very commendable!) it appears that the Wolbachia coverage was taken from the Spades contig file names. It is important to notice that the coverage there does not correspond to whole genome coverage, but rather to kmer coverage of the largest kmer used for assembly (51 in your case). This can be considerably lower than the actual genome coverage.

I suggest for a more exact estimate of Wolbachia genome coverage to map all Wolbachia reads used for assembly back to the Wolbachia assembly, and then use samtools to estimate coverage. If you don’t want to do that (which I’d understand), you could just calculate (number of reads used for assembly)*(read length) / (Wolbachia genome size).

Also, is the Wolbachia coverage normalised to library size? In your calculation for A. gambiae for example, you use all reads to calculate the coverage of the host genome, but it appears that only the dowloaded reads (5x10^7) were used to calculate Wolbachia coverage. If so, this needs to be corrected.

Minor comment: there are two very recent preprint articles that report on Wolbachia in Anopheles that could be cited (https://doi.org/10.1101/338434 ; https://doi.org/10.1101/343715 )

·

Basic reporting

no comment

Experimental design

no comment

Validity of the findings

no comment

Additional comments

This is a fun paper and the authors have done a great job in revising the manuscript. I believe they have addressed all of the reviewers concerns and will be pleased to see it published.

The section in the discussion providing best practice advice, while important, is largely already covered in other publications (see weblink and references below). Im not suggesting removing it, as I believe it is valuable to the community - particularly regards pre-filtering of 'contaminants'. It would be nice to see references to relevant literature for interested readers however.

e.g.

http://fged.org/projects/best-practices-for-omics-data-sharing/
https://www.nature.com/articles/sdata201618
https://f1000research.com/articles/6-1618/v2

---

## Round 0.3 · accepted · Accept

Dear Drs. Pascar and Chandler:

Thanks for again revising your manuscript based on the minor concerns raised by the reviewers. I especially thank you for the effort with reassembling certain genomes. Well done! I now believe that your manuscript is suitable for publication. Congratulations! I look forward to seeing this work in print, and I anticipate it being an important resource for the Wolbachia communiy. Thanks again for choosing PeerJ to publish such important work.

Best,

-joe

#